# Subspace Network: Deep Multi-Task Censored Regression for Modeling Neurodegenerative Diseases

## Abstract

Over the past decade a wide spectrum of machine learning models have been developed to model the neurodegenerative diseases, associating biomarkers, especially non-intrusive neuroimaging markers, with key clinical scores measuring the cognitive status of patients. Multi-task learning (MTL) has been extensively explored in these studies to address challenges associated to high dimensionality and small cohort size. However, most existing MTL approaches are based on linear models and suffer from two major limitations: 1) they cannot explicitly consider upper/lower bounds in these clinical scores; 2) they lack the capability to capture complicated non-linear effects among the variables. In this paper, we propose the *Subspace Network*, an efficient deep modeling approach for non-linear multi-task censored regression. Each layer of the subspace network performs a multi-task censored regression to improve upon the predictions from the last layer via sketching a low-dimensional subspace to perform knowledge transfer among learning tasks. Under mild assumptions, for each layer the parametric subspace can be recovered using only one pass of training data. Empirical results demonstrate that the proposed subspace network quickly picks up correct parameter subspaces, and outperforms state-of-the-arts in predicting neurodegenerative clinical scores using information in brain imaging.

## 1 Introduction

Recent years have witnessed increasing interests on applying machine learning (ML) techniques to analyze biomedical data. Such data-driven approaches deliver promising performance improvements in many challenging predictive problems. For example, in the field of neurodegenerative diseases such as Alzheimer's disease and Parkinson's disease, researchers have exploited ML algorithms to predict the cognitive functionality of the patients from the brain imaging scans, e.g., using the magnetic resonance imaging (MRI) as in Adeli-Mosabbeb et al. (2015); Zhang et al. (2012); Zhou et al. (2011b). As a key finding, there are typically various types of prediction targets (e.g., cognitive scores), and they can be jointly learned using multi-task learning (MTL), e.g., Caruana (1998); Evgeniou & Pontil (2004); Zhang et al. (2012), where the predictive information is shared and transferred among related models to reinforce their generalization performance.

Two challenges persist despite the progress of applying MTL in disease modeling problems. First, it is important to notice that clinical targets, different from typical regression targets, are often naturally bounded. For example, the result from Mini-Mental State Examination (MMSE) test, a key reference for deciding cognitive impairments, ranges from 0 to 30 (a healthy subject): a smaller score indicates a higher level of cognitive dysfunction (please refer to Tombaugh & McIntyre (1992)). Other cognitive scores, such as Clinical Dementia Rating Scale (CDR) Hughes et al. (1982) and Alzheimer's Disease Assessment Scale-Cog (ADAS- Cog) Rosen et al. (1984), also have specific upper and lower bounds. Most existing approaches, e.g., Zhang et al. (2012); Zhou et al. (2011b); Poulin et al. (2011), relied on linear regression without considering the range constraint, partially due to the fact that mainstream MTL models for regression, e.g., Jalali et al. (2010); Argyriou et al. (2007); Zhang et al. (2012); Zhou et al. (2011a), are developed using the least squares loss and cannot be directly extended to censored regressions. As the second challenge, a majority of MTL research focused on linear models because of computational efficiency and theoretical guarantees. However, linear models cannot capture the complicated non-linear relationship between features and clinical

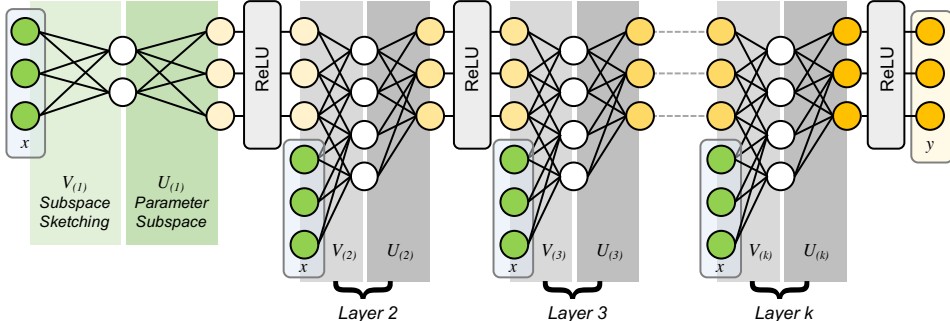

Figure 1: The proposed subspace network via hierarchical subspace sketching and refinement.

targets. For example, Association et al. (2013) showed the early onset of Alzheimer's disease to be related to single-gene mutations on chromosomes *21*, *14*, and *1*, and the effects of such mutations on the cognitive impairment are hardly linear (please refer to Martins et al. (2005); Sweet et al. (2012)). Recent advances in multi-task deep neural networks Seltzer & Droppo (2013); Zhang et al. (2014); Wu et al. (2015) provide a promising direction, but their model complexity and high demands of training data prohibit their broader usages in clinical cohort studies.

To address the aforementioned challenges, we propose a *novel, solid, and efficient* deep modeling approach for non-linear multi-task censored regression, called *Subspace Network* (SN), highlighting the following multi-fold technical innovations:

- It efficiently builds up a deep network in a **layer-by-layer feedforward** fashion, and in each layer considers a **censored** regression problem. The layer-wise training allows us to grow a deep model efficiently.
- It explores a **low-rank** subspace structure that captures task relatedness for better predictions. A critical difference on subspace decoupling between previous studies such as Mardani et al. (2015) Shen et al. (2016) and our method lies in our assumption of a low-rank structure on the parameter space among tasks rather than the original feature space.
- By leveraging the recent advances in online subspace sensing Mardani et al. (2015); Shen et al. (2016), we show that the parametric subspace can be recovered for each layer with feeding **only one pass** of the training data, allowing for more efficient layer-wise training.

Synthetic experiments verify the technical claims of the proposed SN, and it outperforms various state-of-the-arts methods in modeling neurodegenerative diseases on real datasets.

## 2 MULTI-TASK CENSORED REGRESSION VIA PARAMETER SUBSPACE SKETCHING AND REFINEMENT

In censored regression, we are given a set of $N$ observations $\mathcal{D} = \{(x_i, y_i)\}_{i=1}^{N}$ of $D$ dimensional feature vectors $\{x_i \in \mathbb{R}^D\}$ and $T$ corresponding outcomes $\{y_i \in \mathbb{R}_+^T\}$, where each outcome $y_{i,t} \in \mathbb{R}_+$, $t \in \{1, \cdots, T\}$, can be cognitive scores (e.g., MMSE and ADAS-Cog) or other biomarkers of interest such as proteomics[1]. For each outcome, the censored regression assumes a nonlinear relationship between the features and the outcome through a rectified linear unit (ReLU) transformation, i.e., $y_{i,t} = \text{ReLU}\left(W_t^\top x_i + \epsilon\right)$ where $W_t \in \mathbb{R}^D$ is the coefficient for input features, $\epsilon$ is *i.i.d.* noise, and ReLU is defined by $\text{ReLU}(z) = \max(z, 0)$. We can thus collectively represent the censored regression for multiple tasks by:

$$y_i = \text{ReLU}\left(W x_i + \epsilon\right), \tag{1}$$

where $W = [W_1, \ldots, W_T]^\top \in \mathbb{R}^{T \times D}$ is the coefficient matrix. We consider the regression problem for each outcome as a learning task, and one commonly used task relationship assumption is that the transformation matrix $W \in \mathbb{R}^{T \times D}$ belongs to a linear low-rank subspace $\mathcal{U}$. The subspace allows us to represent $W$ as product of two matrices, $W = UV$, where columns of $U \in \mathbb{R}^{T \times R} = [U_1, \ldots, U_T]^\top$ span the linear subspace $\mathcal{U}$, and $V \in \mathbb{R}^{R \times D}$ is the embedding coefficient. We note that the output $y$ can be entry-wise decoupled, such that for each component $y_{i,t} = \text{ReLU}(U_t^\top V x_i + \epsilon)$. By assuming Gaussian noise $\epsilon \sim \mathcal{N}(0, \sigma^2)$, we derive the following likelihood function:

$$\Pr(y_{i,t}, x_i | U_t, V) = \phi\left(\frac{y_{i,t} - U_t^\top V x_i}{\sigma}\right) \mathbb{I}(y_{i,t} \in (0, \infty)) + \left[1 - Q\left(\frac{0 - U_t^\top V x_i}{\sigma}\right)\right] \mathbb{I}(y_{i,t} = 0),$$

---

[1]Without loss of generality, in this paper we assume that outcomes are lower censored at 0. By using variants of Tobit models, e.g., as in Shen et al. (2016), the proposed algorithms and analysis can be extended to other censored models with minor changes in the loss function.

---

**Algorithm 1** Single-layer parameter subspace sketching and refinement.

---

**Require:** Training data $\mathcal{D} = \{(x_i, y_i)\}_{i=1}^N$, rank parameters $\lambda$ and $R$,
**Ensure:** parameter subspace $U$, parameter sketch $V$
  Initialize $U^-$ at random
  **for** $i = 1, \ldots, N$ **do**
    *// 1. Sketching parameters in the current subspace*
$$V^+ = \arg\min_V - \log \Pr(y_i, x_i | U^-, V) + \tfrac{\lambda}{2}\|V\|_F^2$$
    *// 2. Parallel subspace refinement* $\{U_t\}_{t=1}^T$
    **for** $t = 1, \ldots, T$ **do**
      $U_t^+ = \arg\min_{U_t} - \log \Pr(y_{i,t}, x_i | U_t, V^+) + \tfrac{\lambda}{2}\|U_t\|_2^2$
    **end for**
    Set $U^- = U^+, V^- = V^+$
  **end for**

---

where $\phi$ is the probabilistic density function of the standardized Gaussian $N(0,1)$ and $Q$ is the standard Gaussian tail. $\sigma$ controls how accurately the low-rank subspace assumption can fit the data. Note that other noise models can be assumed here too. The likelihood of $(x_i, y_i)$ pair is thus given by:

$$\Pr(y_i, x_i | U, V) = \prod_{t=1}^T \left\{ \phi\left(\frac{y_{i,t} - U_t^\top V x_i}{\sigma}\right) \mathbb{I}(y_{i,t} \in (0, \infty)) + \left[1 - Q\left(-\frac{U_t^\top V x_i}{\sigma}\right)\right] \mathbb{I}(y_{i,t} = 0) \right\}.$$

The likelihood function allows us to estimate subspace $U$ and coefficient $V$ from data $\mathcal{D}$. To enforce a low-rank subspace, one common approach is to impose a trace norm on $UV$, where trace norm of a matrix $A$ is defined by $\|A\|_* = \sum_j s_j(A)$ and $s_j(A)$ is the $j$th singular value of $A$. Since $\|UV\|_* = \min_{U,V} \frac{1}{2}(\|U\|_F^2 + \|V\|_F^2)$, e.g., see Srebro et al. (2005); Mardani et al. (2015), the objective function of multi-task censored regression problem is given by:

$$\min_{U,V} - \sum_{i=1}^N \log \Pr(y_i, x_i | U, V) + \tfrac{\lambda}{2}(\|U\|_F^2 + \|V\|_F^2). \tag{2}$$

### 2.1 An online algorithm

We propose to solve the objective in (2) via the block coordinate descent approach which is reduced to iteratively updating the following two subproblems:

$$V^+ = \arg\min_V - \sum_{i=1}^N \log \Pr(y_i, x_i | U^-, V) + \tfrac{\lambda}{2}\|V\|_F^2, \tag{P:V}$$

$$U^+ = \arg\min_U - \sum_{i=1}^N \log \Pr(y_i, x_i | U, V^+) + \tfrac{\lambda}{2}\|U\|_F^2. \tag{P:U}$$

Define the instantaneous cost of the $i$-th datum:

$$g(x_i, y_i, U, V) = - \log \Pr(x_i, y_i | U, V) + \tfrac{\lambda}{2}\|U\|_F^2 + \tfrac{\lambda}{2}\|V\|_F^2,$$

and the online optimization form of (2) can be recast as an empirical cost minimization given below:

$$\min_{U,V} \frac{1}{N} \sum_{i=1}^N g(x_i, y_i, U, V).$$

According to the analysis in Section 2.2, one pass of the training data can warrant the subspace learning problem. We outline the solver for each subproblem as follows:

**Problem** (P:V) **sketches parameters in the current space.** We solve (P:V) using gradient descent. The parameter sketching couples all the subspace dimensions in $V$ (not decoupled as in Shen et al. (2016)), and thus we need to solve this collectively. The update of $V$ ($V^+$) can be obtained by solving the online problem given below:

$$\min_V g(x_i, y_i; U^-, V) \equiv -\sum_{t=1}^T \log \Pr(y_{i,t}, x | U_t^-, V) + \frac{\lambda}{2}\|V\|_F^2$$

$$= -\sum_{t=1}^T \log \left[ \phi\left(\frac{y_{i,t} - \left(U_t^-\right)^\top V x}{\sigma}\right) \mathbb{I}(y_{i,t} \in (0, \infty)) + \left[1 - Q\left(\frac{-\left(U_t^-\right)^\top V x}{\sigma}\right)\right] \mathbb{I}(y_{i,t} = 0) \right] + \frac{\lambda}{2}\|V\|_F^2.$$

$V^+$ can be computed by the following gradient update: $V^+ = V^- - \eta \nabla_V g(x_i, y_i; U^-, V^+)$, where the gradient is given by:

$$\nabla_V g(x_i, y_i; U^-, V^+) = \lambda V + \sum_{t=1}^{T} \begin{cases} -\frac{y_{i,t} - (U_t^-)^\top V x_i}{\sigma^2} U_t^- x_i^\top & y_{i,t} \in (0, \infty) \\ \frac{\phi(z_t)}{\sigma[1-Q(z_{i,t})]} U_t^- x_i^T & y_{i,t} = 0 \end{cases}$$

where $z_{i,t} = \sigma^{-1}(-(U_t^-)^\top V x)$. The Alg. 3 for solving (P:V) is summarized in the Appendix.

**Problem** (P:U) **refines the subspace $U^+$ based on sketching.** We solve (P:U) using stochastic gradient descent (SGD). We note that the problem is decoupled for different subspace dimensions $t = 1, \ldots, T$ (i.e., rows of $U$). With careful parallel design, this procedure can be done very efficiently. Given a training data point $(x_i, y_i)$, the problem related to the $t$-th subspace basis is:

$$\min_{U_t} g_t(x_i, y_{i,t}; U_t, V^+) \equiv -\log \Pr(y_{i,t}, x_i | U_t, V^+) + \frac{\lambda}{2} \|U_t\|_2^2$$

$$= -\log \left[ \phi \left( \frac{y_{i,t} - U_t^\top V^+ x_i}{\sigma} \right) \mathbb{I}(y_{i,t} \in (0, \infty)) + \left[ 1 - Q \left( \frac{-U_t^\top V^+ x_i}{\sigma} \right) \right] \mathbb{I}(y_{i,t} = 0) \right] + \frac{\lambda}{2} \|U_t\|_2^2.$$

We can revise subspace by the following gradient update: $U_t^+ = U_t^- - \mu_t \nabla_{U_t} g_t(x_i, y_{i,t}; U_t, V^+)$, where the gradient is given by:

$$\nabla_{U_t} g_t(x_i, y_{i,t}; U_{i,t}, V^+) = \lambda U_t + \begin{cases} -\frac{y_{i,t} - U_t^\top V^+ x}{\sigma^2} V^+ x_i & y_{i,t} \in (0, \infty) \\ \frac{\phi(z_{i,t})}{\sigma[1-Q(z_{i,t})]} V^+ x_i & y_{i,t} = 0 \end{cases}$$

where $z_{i,t} = \sigma^{-1}(-U_t^\top V^+ x_i)$. We summarize the procedure in Algorithm 1 and show in Section 2.2 that under mild assumptions this procedure will be able to capture the underlying subspace structure in the parameter space with just one pass of the data.

## 2.2 THEORETICAL RESULTS

We establish both asymptotic and non-asymptotic convergence properties for Algorithm 1. The proof scheme is inspired by a series of previous works: Mairal et al. (2010); Kasiviswanathan et al. (2012); Shalev-Shwartz et al. (2012); Mardani et al. (2013; 2015); Shen et al. (2016). We briefly present the proof sketch, and more proof details can be found in Appendix. At each iteration $i = 1, 2, ..., N$, we sample $(x_i, y_i)$, and let $U^i, V^i$ denote the intermediate $U$ and $V$, to be differentiated from $U_t, V_t$ which are the $t$-th columns of $U, V$. For the proof feasibility, we assume that $\{(x_i, y_i)\}_{i=1}^N$ are sampled i.i.d., and the subspace sequence $\{U^i\}_{i=1}^N$ lies in a compact set.

**Asymptotic Case:** To estimate $U$, the Stochastic Gradient Descent (SGD) iterations can be seen as minimizing the approximate cost $\frac{1}{N} \sum_{i=1}^N g'(x_i, y_i, U, V)$, where $g'$ is a tight quadratic surrogate for $g$ based on the second-order Taylor approximation around $U^{N-1}$. Furthermore, $g$ can be shown to be smooth, by bounding its first-order and second-order gradients w.r.t. each $U_t$ (similar to Appendix 1 of Shen et al. (2016)).

Following Mairal et al. (2010); Mardani et al. (2015), it can then be established that, as $N \to \infty$, *the subspace sequence $\{U^i\}_{i=1}^N$ asymptotically converges to a stationary-point of the batch estimator, under a few mild conditions*. We can sequentially show: 1) $\sum_{i=1}^N g'(x_i, y_i, U^i, V^i)$ asymptotically converges to $\sum_{i=1}^N g(x_i, y_i, U^i, V^i)$, according to the quasi-martingale property in the almost sure sense, owing to the tightness of $g'$; 2) the first point implies convergence of the associated gradient sequence, due to the regularity of $g$; 3) $g_t(x_i, y_i, U, V)$ is bi-convex for block variables $U_t$ and $V$.

**Non-Asymptotic Case:** When $N$ is finite, Mardani et al. (2013) asserts that *the distance between successive subspace estimates will vanish as fast as $o(1/i)$: $\|U^i - U^{i-1}\|_F \leq \frac{B}{i}$, for some constant $B$ that is independent of $i$ and $N$*. Following Shen et al. (2016) to exploit the unsupervised formulation of regret analysis as in Kasiviswanathan et al. (2012); Shalev-Shwartz et al. (2012), we can similarly obtain a tight regret bound that will again vanish if $N \to \infty$.

## 3 SUBSPACE NETWORK VIA HIERARCHICAL SKETCHING AND REFINEMENT

The single layer model in (1) has very limited power to capture the highly nonlinear regression relationships, as the parameters are linearly linked to the subspace except for a ReLU operation.

---

**Algorithm 2** Network expansion via hierarchical parameter subspace sketching and refinement.

---

**Require:** Training data $\mathcal{D} = \{(x_i, y_i)\}$, target network depth $K$.
**Ensure:** The deep subspace network $f$
   Set $f_{[0]}(x) = y$ and solve $f_{[0]}$ using Algorithm 1.
   **for** $k = 1, \ldots, K - 1$ **do**
       *// 1. Subspace sketching based on the current subspace using Algorithm 1*:

$$U_{[k]}^*, V_{[k]}^* = \underset{U_{[k]}, V_{[k]}}{\arg \min} \, \mathbb{E}_{(x,y) \sim \mathcal{D}} \left\{ \ell(y, \mathrm{ReLU}\left(U_{[k]} V_{[k]} f_{[k-1]}(x)\right)) \right\},$$

       *// 2. Expand the layer using the refined subspace as our new network*:

$$f_{[k]}(x) = \mathrm{ReLU}\left(U_{[k]}^* V_{[k]}^* f_{[k-1]}(x)\right)$$

   **end for**
   **return** $f = f_{[K]}$

---

However, the single-layer procedure in Algorithm 1 has provided a building block, based on which we can develop an efficient algorithm to train a deep *subspace network* (SN) in a greedy fashion. We thus propose a network expansion procedure to overcome such limitation.

After we obtain the parameter subspace $U$ and sketch $V$ for the single-layer case (1), we project the data points by $\bar{x} = \mathrm{ReLU}(UVx)$. A straightforward idea of the expansion is to use $(\bar{x}, y)$ as the new samples to train another layer. Let $f_{[k-1]}$ denote the network structure we obtained before the $k$-th expansion starts, $k = 1, 2, ..., K - 1$, the expansion can recursively stack more ReLU layers:

$$f_{[k]}(x) = \mathrm{ReLU}\left(U_{[k]} V_{[k]} f_{[k-1]}(x) + \epsilon\right), \tag{3}$$

However, we observe that simply stacking layers by repeating (3) many times can cause substantial information loss and degrade the generalization performance, especially since our training is layer-by-layer without "looking back" (i.e., top-down joint tuning). Inspired by deep residual networks by He et al. (2016) that exploit "skip connections" to pass lower-level data and features to higher levels, we concatenate the original samples with the newly transformed, censored outputs after each time of expansion, i.e., reformulating $\bar{x} = [\mathrm{ReLU}(UVx); x]$ (similar manners could be found in Zhou & Feng (2017)). The new form after the expansion is given below:

$$f_{[k]}(x) = \mathrm{ReLU}\left(U_{[k]} V_{[k]} [f_{[k-1]}(x); x] + \epsilon\right).$$

We summarize the network expansion process in Algorithm 2. The architecture of the resulting SN is illustrated in Figure 1. Compared to the single layer model (1), SN gradually refines the parameter subspaces by multiple stacked nonlinear projections. It is expected to achieve superior performance due to the higher learning capacity, and can also be viewed as a gradient boosting method. Meanwhile, the layer-wise low-rank subspace structural prior would improve generalization further compared to the naive multi-layer networks.

## 4 EXPERIMENT

### 4.1 SIMULATIONS ON SYNTHETIC DATA

**Subspace recovery in a single layer model.** We first evaluate the subspace recovered by the proposed Algorithm 1 using synthetic data. We generated $X \in \mathbb{R}^{N \times D}$, $U \in \mathbb{R}^{T \times R}$ and $V \in \mathbb{R}^{R \times D}$, all as i.i.d. random Gaussian matrices. The target matrix $Y \in \mathbb{R}^{N \times T}$ was then synthesized using (1). We set $N = 5,000$, $D = 200$, $T = 100$ $R = 10$, and use random noise $\epsilon \sim \mathcal{N}(0, 3^2)$.

Figure 2a shows the plot of *subspace difference* between the ground-truth $U$ and the learned subspace $U_i$ throughout the iterations, i.e., $\|U - U_i\|_F / \|U\|_F$ w.r.t. $i$. This result verifies that Algorithm 1 is able to correctly find and smoothly converge to the underlying low-rank subspace of the synthetic data. The objective values throughout the online training process of Algorithm 1 are plotted in Figure 2b. We further show the plot of *iteration-wise subspace differences*, defined as $\|U_i - U_{i-1}\|_F / \|U\|_F$, in Figure 2c, which complies with the $o(1/t)$ result in our non-asymptotic analysis. Moreover, the

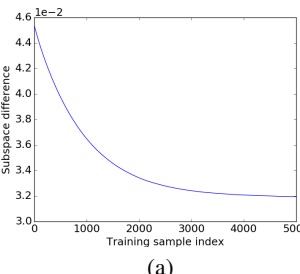 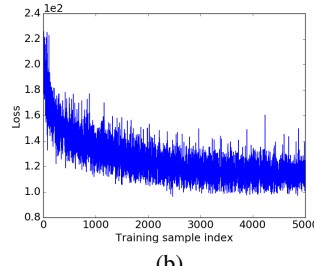 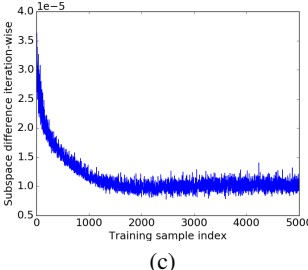

Figure 2: (a) Subspace differences, w.r.t. the index $i$; (b) Convergence of Algorithm 1, w.r.t. the index $i$; (c) Iteration-wise subspace differences, w.r.t. the index $i$.

Table 1: Comparison of subspace differences for each layer of SN, f-MLP, and rf-MLP.

| Metric | Subspace Difference | | | Maximum Mutual Coherence | | | Mean Mutual Coherence | | |
|---|---|---|---|---|---|---|---|---|---|
| Method | SN | f-MLP | rf-MLP | SN | f-MLP | rf-MLP | SN | f-MLP | rf-MLP |
| Layer 1 | **0.0313** | 0.0315 | 0.0317 | 0.7608 | 0.7727 | **0.7895** | **0.2900** | 0.2725 | 0.2735 |
| Layer 2 | 0.0321 | 0.0321 | 0.0321 | **0.8283** | 0.7603 | 0.7654 | **0.2882** | 0.2820 | 0.2829 |
| Layer 3 | **0.0312** | 0.0315 | 0.0313 | **0.8493** | 0.7233 | 0.7890 | **0.2586** | 0.2506 | 0.2485 |

distribution of correlation between recovered weights and true weights for all tasks is given in Figure 3 in Appendix, with most predicted weights having correlations with ground truth of above 0.9.

**Subspace recovery in a multi-layer subspace network.** We re-generated synthetic data by repeatedly applying (1) for three times, each time following the same setting as the single-layer model. A three-layer SN was then learned using Algorithm 2. As one simple baseline, a *multi-layer perceptron* (MLP) is trained, whose three hidden layers have the same dimensions as the three ReLU layers of the SN. Inspired by Xue et al. (2013); Sainath et al. (2013); Wang et al. (2015), we then applied low-rank matrix factorization to each layer of MLP, with the same desired rank $R$, creating the *factorized MLP* (f-MLP) baseline that has the identical architecture (including both ReLU hidden layers and linear bottleneck layers) to SN. We further re-trained the f-MLP on the same data from end to end, leading to another baseline, the *retrained factorized MLP* (rf-MLP).

Table 1 evaluates the subspace recovery fidelity in three layers, using three different metrics: (1) the maximum mutual coherence of all column pairs from two matrices, defined in (Candes & Romberg (2007)) as a classical measurement on how correlated the two matrices' column subspaces are; (2) the mean mutual coherence of all column pairs from two matrices; (3) the subspace difference defined the same as in the single-layer case[2]. Note that the two mutual coherence-based metrics are immune to linear transformations of subspace coordinates, to which the $\ell_2$-based subspace difference might become fragile. SN achieves clear overall advantages under all three measurements, over f-MLP and rf-MLP. More notably, while the performance margin of SN in subspace difference seems to be small, the much sharper margins, in two (more robust) mutual coherence-based measurements, suggest that the recovered subspaces by SN are significantly better aligned with the groundtruth.

**Benefits of Going Deep.** We re-generate synthetic data again in the same way as the first single-layer experiment; yet differently, we now aim to show that a deep SN will boost performance over single-layer subspace recovery, even the data generation does not follow a known multi-layer model. We compare SN (both 1-layer and 3-layer) with two carefully chosen sets of state-of-art approaches: (1) single and multi-task "shallow" models; (2) deep models. For the first set, the least squares (LS) is treated as a naive baseline, while ridge (LS + $\ell_2$) and lasso (LS + $\ell_1$) regressions are considered for shrinkage or variables selection purpose; Censor regression, also known as the Tobit model, is a **non-linear** method to predict bounded targets , e.g., Berberidis et al. (2016). Multi-task models with regularizations on trace norm (Multi Trace) and $\ell_{2,1}$ norm (Multi $\ell_{2,1}$) have been demonstrated to be successful on simultaneous structured/sparse learning, e.g., Yang et al. (2010); Zhang et al. (2013).[3] We also verify the benefits of accounting for boundedness of targets (Uncensored vs. Censored) in both single-task and multi-task settings, with **best** performance reported

---

[2]The higher in terms of the two mutual coherence-based metrics, the better subspace recovery is achieved.That is different from the subspace difference case where the smaller the better,

[3]Least squares, ridge, lasso, and censor regression are implemented by MATLAB optimization toolbox. MTLs are implemented through MALSAR Zhou et al. (2011a) with parameters carefully tuned.

Table 2: Average normalized mean square error under different approaches for synthetic data. Standard deviation of 10 trials is given in parenthesis.

| Percent | Single Task (Shallow) | | | Multi Task (Shallow) | |
|---|---|---|---|---|---|
| | Uncensored (LS + $\ell_1$) | Censored (LS + $\ell_1$) | Nonlinear Censored (Tobit) | Uncensored (Multi Trace) | Censored (Multi Trace) |
| 40% | 0.1412 (0.0007) | **0.1127** (0.0010) | **0.0428** (0.0003) | 0.1333 (0.0009) | **0.1053** (0.0027) |
| 50% | 0.1384 (0.0005) | **0.1102** (0.0010) | **0.0408** (0.0004) | 0.1323 (0.0010) | **0.1054** (0.0042) |
| 60% | 0.1365 (0.0005) | **0.1088** (0.0009) | **0.0395** (0.0003) | 0.1325 (0.0012) | **0.1031** (0.0046) |
| 70% | 0.1349 (0.0005) | **0.1078** (0.0010) | **0.0388** (0.0004) | 0.1315 (0.0013) | **0.1024** (0.0042) |
| 80% | 0.1343 (0.0011) | **0.1070** (0.0012) | **0.0383** (0.0006) | 0.1308 (0.0008) | **0.1040** (0.0011) |

| Percent | Deep Neural Network | | | Subspace Net (SN) | |
|---|---|---|---|---|---|
| | DNN i (naive) | DNN ii (censored) | DNN iii (censored + low-rank) | Layer 1 | Layer 3 |
| 40% | 0.0623 (0.0041) | 0.0489 (0.0035) | **0.0431** (0.0041) | 0.0390 (0.0004) | **0.0369** (0.0002) |
| 50% | 0.0593 (0.0048) | 0.0462 (0.0042) | **0.0400** (0.0039) | 0.0389 (0.0007) | **0.0366** (0.0003) |
| 60% | 0.0587 (0.0053) | 0.0455 (0.0054) | **0.0395** (0.0050) | 0.0388 (0.0006) | **0.0364** (0.0003) |
| 70% | 0.0590 (0.0071) | 0.0447 (0.0043) | **0.0386** (0.0058) | 0.0388 (0.0006) | **0.0363** (0.0003) |
| 80% | 0.0555 (0.0057) | 0.0431 (0.0053) | **0.0380** (0.0057) | 0.0390 (0.0008) | **0.0364** (0.0005) |

Table 3: Average normalized mean square error at each layer for subspace network ($R = 10$) for synthetic data. Standard deviation of 10 trials is given in parenthesis.

| Percent | Layer 1 | Layer 2 | Layer 3 | Layer 10 | Layer 20 |
|---|---|---|---|---|---|
| 40% | 0.0390 (0.0004) | 0.0381 (0.0005) | 0.0369 (0.0002) | 0.0368 (0.0002) | **0.0368** (0.0002) |
| 50% | 0.0389 (0.0007) | 0.0379 (0.0005) | 0.0366 (0.0003) | 0.0366 (0.0003) | **0.0365** (0.0003) |
| 60% | 0.0388 (0.0006) | 0.0378 (0.0004) | 0.0364 (0.0003) | 0.0364 (0.0003) | **0.0363** (0.0003) |
| 70% | 0.0388 (0.0006) | 0.0378 (0.0005) | 0.0363 (0.0003) | 0.0363 (0.0003) | **0.0362** (0.0003) |
| 80% | 0.0390 (0.0008) | 0.0378 (0.0006) | 0.0364 (0.0005) | 0.0363 (0.0005) | **0.0363** (0.0005) |

for each scenario (LS + $\ell_1$ for single-task and Multi Trace for multi-task). For the set of deep model baselines, we construct three DNNs for fair comparison: i) A 3-layer fully connected DNN with the same architecture as SN, with a plain MSE loss; ii) A 3-layer fully connected DNN as i) with ReLU added for output layer before feeding into the MSE loss, which naturally implements non-negativity *censored* training and evaluation; iii) A factorized and re-trained DNN from ii), following the same procedure of rf-MLP in the multi-layer synthetic experiment. Apparently, ii) and iii) are constructed to verify if DNN also benefits from the censored target and the low-rank assumption, respectively.

We performed 10-fold random-sampling validation on the same dataset, i.e., randomly splitting into training and validation data 10 times. For each split, we fit model on training data and evaluated performance on validation data. Average normalized mean square error (ANMSE) across all tasks is obtained as the overall performance for each split. For methods without hyper parameters (least square and censor regression), an average of ANMSE for 10 splits is regarded as the final performance; for methods with tunable parameters, e.g., $\lambda$ in lasso, we perform a grid search on $\lambda$ values and choose the optimal ANMSE result. We consider different splitting size with training samples containing [40%, 50%, 60%, 70%, 80%] of all the samples.

Table 2 further compares the performance of all approaches. We see that: (1) all **censored** models significantly outperform their uncensored counterparts, verifying the necessity of adding censoring targets for regression. Therefore, we will use censored baselines hereinafter, unless otherwise specified; (2) the more structured **MTL** models tend to outperform single task models by capturing task relatedness. That is also evidenced by the performance margin of DNN iii over DNN i; (3) the **nonlinear** models are undoubtedly more favorable: we even see the single-task Tobit model to outperform MTL models; (4) As a *nonlinear, censored MTL model*, SN combines the best of them all, accounting for its superior performance over all competitors. In particular, even a 1-layer SN already produces comparable performance to the 3-layer DNN iii (which also a nonlinear, censored MTL model trained with back-propagation, with three times the parameter amount of SN), thanks to SN's theoretically solid online algorithm in characterizing subspaces.

Furthermore, we grow the number of layers in SN from 2 to 20, finding that as a feed-forward model without end-to-end optimization, SN can also benefit from its increasing depth. As Table 3 reveals, SN steadily improves with more layers, until reaching a plateau at around 5 layers (as the underlying data distribution is relatively simple here). The observation is consistent among different splits.

## 4.2 EXPERIMENTS ON REAL DATA

We evaluated SN in a real clinical setting with the goal to build models to predict important clinical scores representing a subject's cognitive status and signaling the progression of Alzheimer's disease (AD), from structural Magnetic Resonance Imaging (sMRI) data. AD is one major neurodegenerative disease that accounts for 60 to 80 percent of dementia. The National Institutes of Health has thus

Table 5: Average normalized mean square error under different approaches for real data. Standard deviation of 10 trials is given in parenthesis.

| Percent | Single Task (Censored) | | | Multi Task (Censored) | |
|---|---|---|---|---|---|
| | Least Square | LS + $\ell_1$ | Tobit | Multi Trace | Multi $\ell_{2,1}$ |
| 40% | 0.3874 (0.0203) | **0.2393** (0.0056) | 0.3870 (0.0306) | 0.2572 (0.0156) | **0.2006** (0.0099) |
| 50% | 0.3119 (0.0124) | **0.2202** (0.0049) | 0.3072 (0.0144) | 0.2406 (0.0175) | **0.2002** (0.0132) |
| 60% | 0.2779 (0.0123) | **0.2112** (0.0055) | 0.2719 (0.0114) | 0.2596 (0.0233) | **0.2072** (0.0204) |
| 70% | 0.2563 (0.0108) | **0.2037** (0.0042) | 0.2516 (0.0108) | 0.2368 (0.0362) | **0.2017** (0.0116) |
| 80% | 0.2422 (0.0112) | **0.2005** (0.0054) | 0.2384 (0.0099) | 0.2176 (0.0171) | **0.2009** (0.0050) |

| Percent | Deep Neural Network | | | Subspace Net (SN) | |
|---|---|---|---|---|---|
| | DNN i (naive) | DNN ii (censored) | DNN iii (censored + low-rank) | Layer 1 | Layer 3 |
| 40% | 0.2549 (0.0442) | 0.2388 (0.0121) | **0.2113** (0.0063) | 0.2016 (0.0057) | **0.1981** (0.0031) |
| 50% | 0.2236 (0.0066) | 0.2208 (0.0062) | **0.2127** (0.0118) | 0.1992 (0.0040) | **0.1971** (0.0038) |
| 60% | 0.2215 (0.0076) | 0.2200 (0.0076) | **0.2087** (0.0102) | 0.1990 (0.0061) | **0.1967** (0.0038) |
| 70% | 0.2149 (0.0077) | 0.2141 (0.0079) | **0.2093** (0.0137) | 0.1981 (0.0046) | **0.1953** (0.0039) |
| 80% | 0.2132 (0.0138) | 0.2090 (0.0079) | **0.2069** (0.0135) | 0.1970 (0.0034) | **0.1956** (0.0040) |

focused on studies investigating brain and fluid biomarkes of the disease, and supported the long running project called Alzheimer's Disease Neuroimaging Initiative (ADNI) from 2003. We used the ADNI-1 cohort (http://adni.loni.usc.edu/). In the experiments, we used the 1.5 Tesla structural MRI collected at the baseline, and performed cortical reconstruction and volumetric segmentations with the FreeSurfer following the procotol in Jack et al. (2008). For each MRI image, we extracted 138 features representing the cortical thickness and surface areas of region-of-interests (ROIs) using the Desikan-Killiany cortical atlas Desikan et al. (2006). After preprocessing, we obtained a dataset containing 670 samples and 138 features. These imaging features were used to predict a set of 30 clinical scores including ADAS scores Rosen et al. (1984) at baseline and future (6 months from baseline), baseline Logical Memory from Wechsler Memory Scale IV Scale—Fourth (2009), Neurobattery scores (i.e. immediate recall total score and Rey Auditory Verbal Learning Test scores), and the Neuropsychiatric Inventory Cummings (1997) at baseline and future.

**Calibration.** In MTL formulations we typically assume that noise variance $\sigma^2$ is the same across all tasks, which may not be true in many cases. To deal with heterogeneous $\sigma^2$ among tasks, we design a *calibration* step in our optimization process, where we estimate task-specific $\hat{\sigma}_t^2$ using $\|y - \hat{y}\|_2^2 / N$ before ReLU, as the input for next layer and repeat on layer-wise. We compare performance of both non-calibrated and calibrated methods.

Table 4: ANMSE for non-calibrated vs. calibrated SN for real data (6 layers). Standard deviation of 10 trials is given in parenthesis.

| Percent | Non-calibrate | Calibrate |
|---|---|---|
| 40% | 0.1993 (0.0034) | **0.1977** (0.0031) |
| 50% | 0.1987 (0.0043) | **0.1967** (0.0036) |
| 60% | 0.1991 (0.0044) | **0.1964** (0.0039) |
| 70% | 0.1982 (0.0042) | **0.1951** (0.0038) |
| 80% | 0.1984 (0.0041) | **0.1954** (0.0039) |

**Performance.** We adopt the same two sets of comparison baselines for the real data experiments, as the last synthetic experiment. Different from synthetic data where the low-rank structure is predefined, for real data, there is no groundtruth rank available and we have to try different rank assumptions. Table 4 compares the performances between $\sigma^2$ non-calibrated versus calibrated models. We observe a clear improvement by assuming different $\sigma^2$ across tasks. Table 5 shows the results for all comparison methods, with SN outperforming all else. Table 7 in Appendix shows the SN performance growth with increasing the number of layers. Table 8 in Appendix further reveals the performance of DNNs and SN using varying rank estimations in real data. As expected, the U-shape curve suggests that an overly low rank may not be informative enough to recover the original weight space, while a high-rank structure cannot enforce as strong a structural prior. However, the overall robustness of SN to rank assumptions is fairly remarkable: its performance under all ranks is competitive, consistently outperforming DNNs under the same rank assumptions and other baselines.

## 5   CONCLUSIONS AND FUTURE WORK

In this paper we proposed a *Subspace Network* (SN), an efficient deep modeling approach for non-linear multi-task censored regression, where each layer of the subspace network performs a multi-task censored regression to improve upon the predictions from the last layer via sketching a low-dimensional subspace to perform knowledge transfer among learning tasks. We show that under mild assumptions, for each layer we can recover the parametric subspace using only one pass of training data. We demonstrate empirically that the subspace network can quickly capture correct parameter subspaces, and outperforms state-of-the-arts in predicting neurodegenerative clinical scores from brain imaging. Based on similar formulations, the proposed method can be easily extended to cases where the targets have nonzero bounds, or both lower and upper bounds.

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

# Appendix

We hereby give more proof details of both asymptotic and non-asymptotic convergence properties for Algorithm 1 to recover the latent subspace $U$. The proofs heavily rely on a series of previous results in Mairal et al. (2010); Kasiviswanathan et al. (2012); Shalev-Shwartz et al. (2012); Mardani et al. (2013; 2015); Shen et al. (2016), and many key results are directly referred to hereinafter for conciseness. We include the proofs for the manuscript to be self-contained.

At iteration $i = 1, 2, ..., N$, we sample $(x_i, y_i)$, and let $U^i, V^i$ denote the intermediate $U$ and $V$, to be differentiated from $U_t, V_t$ which are the $t$-th columns of $U, V$. For the proof feasibility, we assume that $\{(x_i, y_i)\}_{i=1}^N$ are sampled i.i.d., and the subspace sequence $\{U^i\}_{i=1}^N$ lies in a compact set.

## PROOF OF ASYMPTOTIC PROPERTIES

For *infinite* data streams with $N \to \infty$, we recall the instantaneous cost of the $i$-th datum:

$$g_i(x_i, y_i, U, V) = -\log \Pr(x_i, y_i | U, V) + \frac{\lambda}{2} \|U\|_F^2 + \frac{\lambda}{2} \|V\|_F^2,$$

and the online optimization form recasted as an empirical cost minimization:

$$\min_U \frac{1}{N} \sum_{i=1}^N g_i(x_i, y_i, U, V).$$

The Stochastic Gradient Descent (SGD) iterations can be seen as minimizing the approximate cost:

$$\min_U \frac{1}{N} \sum_{i=1}^N g_i'(x_i, y_i, U, V).$$

where $g_N'$ is a tight quadratic surrogate for $g_N$ based on the second-order Taylor approximation around $U^{N-1}$:

$$g_N'(x_N, y_N, U, V) = g_N(x_N, y_N, U^{N-1}, V)$$
$$+ \langle \nabla_U g_N(x_N, y_N, U^{N-1}, V), U - U^{N-1} \rangle + \frac{\alpha_N}{2} \|U - U^{N-1}\|_F^2,$$

with $\alpha_N \geq \|\nabla_U^2 g_N(x_N, y_N, U^{N-1}, V)\|$. $g_N'$ is further recognized as a locally tight upper-bound surrogate for $g_N$, with locally tight gradients. Following the Appendix 1 of Shen et al. (2016), we can show that $g_N$ is smooth, with its first-order and second-order gradients bounded w.r.t. each $U_N$.

With the above results, the convergence of subspace iterates can be proven in the same regime developed in Mardani et al. (2015), whose main inspirations came from Mairal et al. (2010) that established convergence of an online dictionary learning algorithm using the martingale sequence theory. In a nutshell, the proof procedure proceeds by first showing that $\sum_{i=1}^N g_i'(x_i, y_i, U^i, V^i)$ asymptotically converges to $\sum_{i=1}^N g_i(x_i, y_i, U^i, V^i)$, according to the quasi-martingale property in the almost sure sense, owing to the tightness of $g'$. It then implies convergence of the associated gradient sequence, due to the regularity of $g$.

Meanwhile, we notice that $g_i(x_i, y_i, U, V)$ is bi-convex for the block variables $U_t$ and $V$ (see Lemma 2 of Shen et al. (2016)). Therefore due to the convexity of $g_N$ w.r.t. $V$ when $U = U^{N-1}$ is fixed, the parameter sketches $V$ can also be updated exactly per iteration.

All above combined, we can claim the asymptotic convergence for the iterations of Algorithm 1: as $N \to \infty$, *the subspace sequence $\{U^i\}_{i=1}^N$ asymptotically converges to a stationary-point of the batch estimator*, under a few mild conditions.

## PROOF OF NON-ASYMPTOTIC PROPERTIES

For *finite* data streams, we rely on the unsupervised formulation of regret analysis Kasiviswanathan et al. (2012); Shalev-Shwartz et al. (2012) to assess the performance of online iterates. Specifically, at

iteration $t$ ($t \leq N$), we use the previous $U^{t-1}$ to span the partial data at $i = 1, 2, ..., t$. Prompted by the alternating nature of iterations, we adopt a variant of the unsupervised regret to assess the goodness of online subspace estimates in representing the partially available data. With $g_t(x_t, y_t, U^{t-1}, V)$ being the loss incurred by the estimate $U^{t-1}$ for predicting the $t$-th datum, the cumulative online loss for a stream of size $T$ is given by:

$$\bar{C}_T := \frac{1}{T} \sum_{\tau=1}^{T} g_\tau(x_\tau, y_\tau, U^{\tau-1}, V). \tag{4}$$

Further, we will assess the cost of the last estimate $U^T$ using:

$$\hat{C}_T = \frac{1}{T} \sum_{\tau=1}^{T} g_\tau(x_\tau, y_\tau, U^T, V). \tag{5}$$

We define $C_T$ as the batch estimator cost. For the sequence $\{U^t\}_{t=1}^{T}$, we define the online regret:

$$\mathcal{R}_T := \hat{C}_T - \bar{C}_T. \tag{6}$$

We investigate the convergence rate of the sequence $\{\mathcal{R}_T\}$ to zero as $T$ grows. Due to the non-convexity of the online subspace iterates, it is challenging to directly analyze how fast the online cumulative loss $\bar{C}_t$ approaches the optimal batch cost $C_t$. As Shen et al. (2016) advocates, we instead investigate whether $\hat{C}_t$ converges to $\bar{C}_t$. That is established by first referring to the Lemma 2 of Mardani et al. (2013): *the distance between successive subspace estimates will vanish as fast as $o(1/t)$*: $\|U^t - U^{t-1}\|_F \leq \frac{B}{t}$, *for some constant $B$ that is independent of $t$ and $N$.*

Next, following the proof of Proposition 2 in Shen et al. (2016), we can similarly show that: if $\{U^t\}_{t=1}^{T}$ and $\{V_t x_t\}_{t=1}^{T}$ are uniformly bounded, i.e., $\|U^t\|_F \leq B_1$, and $\|V_t x_t\|_2 \leq B_2$, $\forall t \leq T$, then with constants $B_1, B_2 > 0$ and by choosing a constant step size $\mu_t = \mu$, we have a bounded regret as:

$$\mathcal{R}_T \leq \frac{B^2(\ln(T) + 1)^2}{2\mu T} + \frac{5B^2}{6\mu T}.$$

---

**Algorithm 3** Gradient descent algorithm for problem P:V.

---

**Require:** Training data $(x_i, y_i)$, $U^-$, step size $\eta$,
**Ensure:** sketch $V$
    Initialize $V^-$ at random.
    *// 1. Perform gradient step and update the current solution of V.*
    **for** $t = 1, \ldots, T$ **do**
        Compute $z_{i,t} = \sigma^{-1}(-\left(U_t^-\right)^\top V x_i)$.
        Compute the gradient for $y_t$:

$$\nabla g_t(x_i, y_{i,t}; U^-, V^+) = \begin{cases} -\frac{y_{i,t} - \left(U_t^-\right)^\top V x_i}{\sigma^2} U_t^- x_i^\top & y_{i,t} \in (0, \infty) \\ \frac{\phi(z_{i,t})}{\sigma[1 - Q(z_{i,t})]} U_t^- x_i^\top & y_{i,t} = 0 \end{cases}$$

    **end for**
    *// 2. Update the current sketch $V^-$*

$$V^+ = V^- - \eta \left[ \sum_{t=1}^{T} \nabla g_t(x, y_t; U^-, V^+) + \lambda V \right]$$

    Set $V^- = V^+$

---

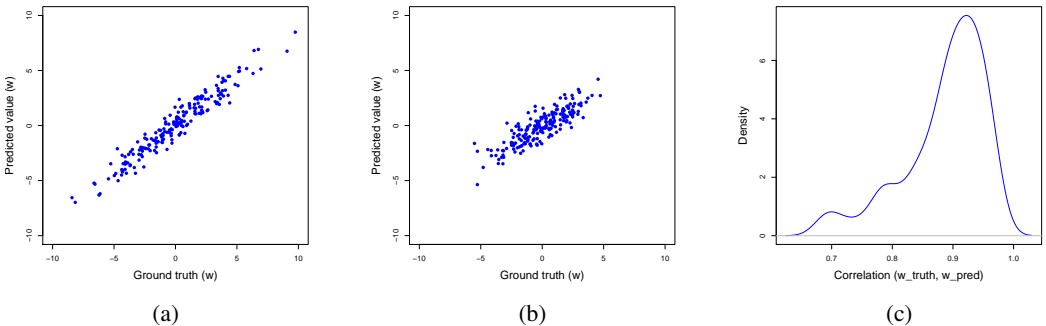

Figure 3: (a) Predicted weight vs true weight for task 1; (b) Predicted weight vs true weight for task 2; (c) Distribution of correlation between predicted weight and true weight for all tasks

## ADDITIONAL EXPERIMENTAL RESULTS

**Computation speed.** All experiments run on the same machine (1 x Six-core Intel Xeon E5-1650 v3 [3.50GHz], 12 logic cores, 128 GB RAM). GPU accelerations are enabled for DNN baselines, while **SN has not exploited the same accelerations yet**. The running time for a single round training on synthetic data (N=5000, D=200, T=100) is given in Table 6. Training each layer of SN will cost 109 seconds on average. As we can see, SN improves generalization performance without significant loss of speed. Furthermore, we will explore to accelerate SN further, by reading data in batch mode and performing parallel updates.

Table 6: Running time and platforms for different methods for synthetic data.

| Method | Time (s) | Platform |
|---|---|---|
| Least Square | 0.02 | Matlab |
| LS+$\ell_2$ | 0.02 | Matlab |
| LS+$\ell_1$ | 18.4 | Matlab |
| Multi-trace | 32.3 | Matlab |
| Multi-$\ell_{21}$ | 27.0 | Matlab |
| Censor | 1680 | Matlab |
| SN (per layer) | 109 | Python |
| DNN | 659 | Tensorflow |

Table 7: Average normalized mean square error at each layer for subspace network ($R = 5$) for real data. Standard deviation of 10 trials is given in parenthesis.

| Percent | Layer 1 | Layer 2 | Layer 3 | Layer 5 | Layer 10 |
|---|---|---|---|---|---|
| 40% | 0.2016 (0.0057) | 0.2000 (0.0039) | 0.1981 (0.0031) | 0.1977 (0.0031) | **0.1977** (0.0031) |
| 50% | 0.1992 (0.0040) | 0.1992 (0.0053) | 0.1971 (0.0038) | 0.1968 (0.0036) | **0.1967** (0.0035) |
| 60% | 0.1990 (0.0061) | 0.1990 (0.0047) | 0.1967 (0.0038) | 0.1964 (0.0039) | **0.1964** (0.0038) |
| 70% | 0.1981 (0.0046) | 0.1966 (0.0052) | 0.1953 (0.0039) | 0.1952 (0.0039) | **0.1951** (0.0038) |
| 80% | 0.1970 (0.0034) | 0.1967 (0.0044) | 0.1956 (0.0040) | 0.1955 (0.0039) | **0.1953** (0.0039) |

Table 8: Average normalized mean square error under different rank assumptions for real data. Standard deviation of 10 trials is given in parenthesis.

| Method | Percent - Rank | $R = 1$ | $R = 3$ | $R = 5$ | $R = 10$ |
|---|---|---|---|---|---|
| SN | 40% | 0.2052 (0.0030) | 0.1993 (0.0036) | **0.1983** (0.0033) | 0.2010 (0.0044) |
| | 50% | 0.2047 (0.0029) | 0.1983 (0.0034) | **0.1981** (0.0043) | 0.2001 (0.0046) |
| | 60% | 0.2052 (0.0033) | 0.1988 (0.0047) | **0.1978** (0.0044) | 0.1996 (0.0052) |
| | 70% | 0.2043 (0.0044) | 0.1975 (0.0042) | **0.1966** (0.0038) | 0.1990 (0.0057) |
| | 80% | 0.2058 (0.0051) | 0.1977 (0.0042) | **0.1965** (0.0046) | 0.1990 (0.0058) |
| DNN iii (censored + low-rank) | 40% | 0.2322 (0.0146) | 0.2360 (0.0060) | **0.2113** (0.0063) | 0.2196 (0.0124) |
| | 50% | 0.2298 (0.0093) | 0.2256 (0.0127) | **0.2127** (0.0118) | 0.2235 (0.0142) |
| | 60% | 0.2244 (0.0132) | 0.2277 (0.0099) | **0.2087** (0.0102) | 0.2145 (0.0208) |
| | 70% | 0.2178 (0.0129) | 0.2177 (0.0115) | **0.2093** (0.0137) | 0.2083 (0.0127) |
| | 80% | 0.2256 (0.0117) | 0.2250 (0.0079) | **0.2069** (0.0135) | 0.2158 (0.0183) |

