# OpenReview forum: "Subspace Network: Deep Multi-Task Censored Regression for Modeling Neurodegenerative Diseases"
_ICLR.cc/2018/Conference — Reject_

### Official Review · AnonReviewer2 · 2017-11-23
**Interesting idea which is however not clearly developed. Incremental results.**

**Rating:** 4
**Confidence:** 5

**Review:**

This work proposes a multi task learning framework for the modeling of clinical data in neurodegenerative diseases.
Differently from previous applications of machine learning in neurodegeneration modeling, the proposed approach models the clinical data accounting for the bounded nature of cognitive tests scores. The framework is represented by a feed-forward deep architecture analogous to a residual network. At each layer a low-rank constraint is enforced on the linear transformation, while the cost function is specified in order to differentially account for the bounds of the predicted variables.

The idea of explicitly accounting for the boundedness of clinical scores is interesting, although the assumption of the proposed model is still incorrect: clinical scores are defined on discrete scales. For this reason the Gaussian assumption for the cost function used in the method is still not appropriate for the proposed application.
Furthermore, while being the main methodological drive of this work, the paper does not show evidence about improved predictive performance and generalisation when accounting for the boundedness of the regression targets.
The proposed algorithm is also generally compared with respect to linear methods, and the authors could have provided a more rigorous benchmark including standard non-linear prediction approaches (e.g. random forests, NN, GP, …).

Overall, the proposed methods seems to provide little added value to the large amount of predictive methods proposed so far for prediction in neurodegenerative disorders. Moreover, the proposed experimental paradigm appears flawed. What is the interest of predicting baseline (or 6 months at best) cognitive scores (relatively low-cost and part of any routine clinical assessment) from brain imaging data (high-cost and not routine)?

Other remarks.

- In section 2.2 and 4 there is some confusion between iteration indices and samples indices “i”.

- Contrarily to what is stated in the introduction, the loss functions proposed in page 3 (first two formulas) only accounts for the lower bound of the predicted variables.

-  Figure 2, synthetic data. The scale of the improvement of the subspace difference is quite tiny, in the order of 1e-2 when compared to U, and of 1e-5 across iterations. The loss function of Figure 2.b also does not show a strong improvement across iterations, while indicating a rather large instability of the optimisation procedure. These aspects may be a sign of convergence issues.

- The dimensionality of the subspace representation importantly depends on the choice of the rank R of U and V. This is a crucial parameters that is however not discussed nor analysed in the paper.

- The synthetic example of page 7 is quite misleading and potentially biased towards the proposed model. The authors are generating the synthetic data according to the model, and it is thus not surprising that they managed to obtain the best performance.  In particular, due to the nonlinear nature of (1), all the competing linear models are expected to perform poorly in this kind of setting.

- The computation time for the linear model shown in Table 3 is quite surprising (~20 minutes for linear regression of 5k observations). Is there anything that I am missing?

---

> ### Author Response · Authors · 2017-12-26
> **Response for Reviewer 3 (Part 1)**
>
> Thanks very much for your suggestions. We have addressed all your concerns below.  Most notably, we have significantly enriched our experiments to testify the performance advantages of SN over DNNs in real data, and to analyze how much each component (censored, low-rank, and online feed-forward training) accounts for this performance gain. We have also revised our paper and corrected typos. Overall, we hope that our revised version has manifested its value as a unique and effective predictive method of neurodegenerative disorders.
>
> Q1: Clinical scores are defined on discrete scales: the Gaussian assumption for the cost function is thus not appropriate for the proposed application.
>
> A1:  As defined in (1), the Gaussian assumption is not enforced on the scores, but on the noise \epsilon that we assume in the latent space. It is a standard assumption in subspace sensing, and the standard deviation hyperparameter controls how accurately the low-rank subspace assumption can fit the data. It does not solely determine the distribution of y, and other noise models can also be assumed here. We apologize for the confusion and have revised the paper to make it clearer.
>
> Q2: The paper does not show evidence about improved predictive performance and generalization when accounting for the target boundedness
>
> A2: We extended experiments to compare results between considering target boundedness (Censored) and not (Uncensored) for both single-task and multi-task models. In either scenario, we reported the best performance (LS+L1 for single task and Multi Trace for multi-task in our case) in Table 2. In the revised version, we further compared SN with several DNN baselines, where the benefit of setting censored regression goals is also found to be evident in DNN settings. Please see next response for details.
>
> Q3: The authors should provide a more rigorous benchmark including non-linear prediction approaches
>
> A3: As suggested, we compared with three DNN baselines (naive, with censoring, and with censoring + low-rank) for both synthetic and real data in the paragraph “Benefits of Going deep” in Section 4.1 and “Performance” in Section 4.2 of the revised paper, in addition to the existing nonlinear Tobit censored regression model. The comparisons of three baselines indicate that both censored regression and low-rank assumption improve DNN’s performance on the given MTL task. Meanwhile, SN clearly outperforms all three, even DNN equipped with censoring + low-rank, suggesting the advantage of our proposed online one-pass sensing and feed-forward training strategy. The performance advantage of SN over DNNs is also confirmed across different rank assumptions in Table 8 in Appendix.
>
> Q4: What is the interest of predicting baseline (or 6 months at best) cognitive scores from brain imaging data?
>
> A4: Thanks for pointing out. We have revised the paper to refer readers interested in this setting to relevant clinical references. The predictive modeling paradigm that we used in the paper is a rather common setting in clinical studies of neurodegenerative diseases such as Alzheimer’s disease (AD), e.g.,
>
> [1] Stonnington, C. M., Chu, C., Klöppel, S., Jack, C. R., Ashburner, J., Frackowiak, R. S., & Alzheimer Disease Neuroimaging Initiative. (2010). Predicting clinical scores from magnetic resonance scans in Alzheimer's disease. Neuroimage, 51(4), 1405-1413.
> [2] Orrù, G., Pettersson-Yeo, W., Marquand, A. F., Sartori, G., & Mechelli, A. (2012). Using support vector machine to identify imaging biomarkers of neurological and psychiatric disease: a critical review. Neuroscience & Biobehavioral Reviews, 36(4), 1140-1152.
> [3] Zhang, D., Shen, D., & Alzheimer's Disease Neuroimaging Initiative. (2012). Multi-modal multi-task learning for joint prediction of multiple regression and classification variables in Alzheimer's disease. NeuroImage, 59(2), 895-907.
> [4] Zhou, J., Liu, J., Narayan, V. A., Ye, J., & Alzheimer's Disease Neuroimaging Initiative. (2013). Modeling disease progression via multi-task learning. NeuroImage, 78, 233-248.
>
> The rationale behind this setting is as follows. For example, the diagnosis of AD requires autopsy confirmation, which is not applicable on live patients. Hence many cognitive measures have been designed to evaluate the cognitive status of a patient. These measures are important criteria for clinical diagnosis of probable AD.  These cognitive status/scores can be considered as phenotypes that are tangled with complicated neurological pathologies in the brain. Currently there are many hypotheses the pathological pathways of AD progression over time, but we are far from understanding the ultimate cause and thus the studies of associations between cognitive scores and neuroimages are critical in understanding the progression and predictability of the disease. The models can reveal important insights and may lead to novel target for therapeutic intervention and drug developments.

---

> > ### Author Response · Authors · 2017-12-26
> > **Response for Reviewer 3 (Part 2)**
> >
> >
> > Q5:  In section 2.2 and 4 there is some confusion between iteration indices and samples indices
> >
> > A5: In our online algorithm, since each iteration a new sample comes in, the iteration indices and samples indices are identical. We apologize for the confusion caused and have clarified it in the revised paper.
> >
> > Q6: Contrarily to what is stated in the introduction, the loss functions proposed in page 3 (first two formulas) only accounts for the lower bound of the predicted variables.
> >
> > A6: Thanks for pointing out. This paper discussed the most basic nonnegativity-censored regression, which coincides with a RELU-based form (1). The effectiveness of this simplest lower bound has been verified in experiments.
> >
> > However, there is no obstacle to extend the developed methodology to general Tobit models with both lower and upper bounds, and to multiple regression targets where each target has varying bounds: that will only affect the form of likelihood function in Section 2. We will certainly extend Subspace Network to account for those cases in future work.
> >
> > Q7: Figure 2, the scale of the improvement of the subspace difference is quite tiny. The loss function of Figure 2.b also does not show a strong improvement across iterations, while indicating a rather large instability of the optimization.
> >
> > A7: (1) Subspace difference is defined as ||U-U_i||_F / ||U||_F, which is divided by the Frobenius norm of U matrix after getting the difference, therefore, it is in small scale; (2) Iteration-wise difference is defined as ||U_i - U_i-1||_F / ||U||_F, similar with (1); (3) our loss is plotted per each single sample passing through the online algorithm: please note the important difference with typical DNN training curves, where loss values are plotted per batch or even per epoch and thus the curve looks smoother. Our training loss, subspace difference between groundtruth, and the iteration-wise subspace differences all steadily decrease as more data points are fed in, showing healthy convergence and fitting our theoretical results well.
> >
> > In the revised version, please further refer to the two new, more robust mutual coherence-based metrics introduced in Section 4.1, in terms of which much sharper margins are achieved by Subspace Network, suggesting that our recovered subspaces are significantly better aligned with the groundtruth.
> >
> > Q8. The dimensionality of the subspace representation importantly depends on the choice of the rank R of U and V. This is not discussed nor analyzed.
> >
> > A8: For real data, since there is no known ground truth of the rank, we choose the rank R by selecting one that leads to overall best performance in cross-validation. We display the effect of rank assumption on Subspace Network performance in real data, in Table 8 of Appendix. The overall robustness of SN to rank assumptions is fairly remarkable: the performance of SN under all rank assumptions appears to be competitive, consistently outperforming DNNs under the same rank assumptions and other baselines.
> >
> > Q9: The synthetic example of page 7 is potentially biased towards the proposed model. The authors are generating the synthetic data according to the model, and it is thus not surprising that they managed to obtain the best performance.
> >
> > A9: One goal of the synthetic experiments is to verify if our model could correctly recover the underlying low rank parameter subspaces: that is why we generate data in the controlled way. The synthetic results align with our theory. Note that the third synthetic experiment in Section 4.1 (“Benefits of Going Deep”) shows an interesting example, that a multi-layer SN performs the best even the data is generated using the one-layer model.
> >
> > More importantly, the practical effectiveness of Subspace Network is validated by the real data experiments, where no data generation process is assumed and no underlying parameter (e.g., rank, layer number) is pre-known. Subspace Network proves to automatically discover latent low-rank subspaces from data and achieves superior predictions. We also compare our model with several DNN baselines in the revised paper, and still achieverperformance margins over them.
> >
> > Q10: The computation time for the linear model shown in Table 3 is quite surprising (~20 minutes for linear regression of 5k observations).
> >
> > A10: Thanks very much for pointing out. There was some unintentional bug in measuring the algorithm running time. We realized and corrected it right after submission. We apologize for this careless mistake and have reported the correct running time in the revised version, Table 6 in Appendix.

---

### Official Review · AnonReviewer1 · 2017-11-26
**The authors propose a DNN, called subspace network, for nonlinear multi-task censored regression problem. The writing needs more elaboration. The experiments are unconvincing.**

**Rating:** 5
**Confidence:** 3

**Review:**

The authors propose a DNN, called subspace network, for nonlinear multi-task censored regression problem. The topic is important. Experiments on real data show improvements compared to several traditional approaches.

My major concerns are as follows.

1. The paper is not self-contained. The authors claim that they establish both asymptotic and non-asymptotic convergence properties for Algorithm 1. However, for some key steps in the proof, they refer to other references. If this is due to space limitation in the main text, they may want to provide a complete proof in the appendix.

2. The experiments are unconvincing. They compare the proposed SN with other traditional approaches on a very small data  set with 670 samples and 138 features. A major merit of DNN is that it can automatically extract useful features. However, in this experiment, the features are handcrafted before they are fed into the models. Thus, I would like to see a comparison between SN with vanilla DNN.

---

> ### Author Response · Authors · 2017-12-26
> **Response for Reviewer 2**
>
>
> Thanks very much for your suggestions. We have addressed all your concerns below.
>
> Q1. The paper is not self-contained. The authors claim that they establish both asymptotic and non-asymptotic convergence properties for Algorithm 1. However, for some key steps in the proof, they refer to other references. If this is due to space limitation in the main text, they may want to provide a complete proof in the appendix.
>
> A1: Thanks very much for your comment. In Appendix of the revised paper, please find the  more detailed long version of the proof outlines of both asymptotic and non-asymptotic convergence properties in Appendix. We expect that the new proof has better self-containedness.
>
> At some steps of the proof, we pointed to the important key results to refer to (in precise forms, e.g., Lemma 2 of Mardani et al. (2013) ). In order to focus on the key contribution of this paper, we did not include the detailed assumptions and proofs of all those intermediate theorems/lemma that we used. The reasons are: (1) they can be very lengthy; (2) they were well-established results in other relevant literature and were not our innovations; (3) those intermediate results were not tightly related the main contributions of this paper (SN model). We believe that the current proof outline has already captured all main proof ideas and should be easy to follow for readers of interests.
>
> Q2. The experiments are unconvincing. They compare the proposed SN with other traditional approaches on a very small data  set with 670 samples and 138 features. A major merit of DNN is that it can automatically extract useful features. However, in this experiment, the features are handcrafted before they are fed into the models. Thus, I would like to see a comparison between SN with vanilla DNN.
>
> A2: Thanks for your comments. We agree that one major merit of DNN is to automatically extract features from images, that demonstrated huge success in many domains. Such capability is based on the availability of large labeled training data. In the medical research domain, however, such labeled data is rarely available, especially in the challenging disease of neurodegenerative diseases such as Alzheimer’s disease (AD) and Parkinson. The ADNI data used in our paper is so far the largest cohort collected for Alzheimer’s disease study, and however has less than 1000 patients available for building predictive models due to the expensive data collection process. Due to extreme high dimension of an MRI image (voxel size: 512x512x16 = 4,194,304), most studies use region-of-interests features extracted by existing neuroimaging tools, instead of raw imaging data for studying progression of a disease. As such, a majority amount of AD study performs predictive modeling using extracted features:
>
> [1] Duchesne, S., Caroli, A., Geroldi, C., Collins, D. L., & Frisoni, G. B. (2009). Relating one-year cognitive change in mild cognitive impairment to baseline MRI features. Neuroimage, 47(4), 1363-1370.
> [2] Stonnington, C. M., Chu, C., Klöppel, S., Jack, C. R., Ashburner, J., Frackowiak, R. S., & Alzheimer Disease Neuroimaging Initiative. (2010). Predicting clinical scores from magnetic resonance scans in Alzheimer's disease. Neuroimage, 51(4), 1405-1413.
> [3] Orrù, G., Pettersson-Yeo, W., Marquand, A. F., Sartori, G., & Mechelli, A. (2012). Using support vector machine to identify imaging biomarkers of neurological and psychiatric disease: a critical review. Neuroscience & Biobehavioral Reviews, 36(4), 1140-1152.
> [4] Zhang, D., Shen, D., & Alzheimer's Disease Neuroimaging Initiative. (2012). Multi-modal multi-task learning for joint prediction of multiple regression and classification variables in Alzheimer's disease. NeuroImage, 59(2), 895-907.
> [5]  Zhou, J., Liu, J., Narayan, V. A., Ye, J., & Alzheimer's Disease Neuroimaging Initiative. (2013). Modeling disease progression via multi-task learning. NeuroImage, 78, 233-248.
>
> We also improve our experiments by comparing with three DNN baselines (naive, with censoring, and with censoring + low-rank) in both synthetic and real data, please refer to the new paragraph “Benefits of Going deep” in Section 4.1 and “Performance” in Section 4.2 in the revised paper. The comparisons of three baselines indicate that both censored regression and low-rank assumption improve DNN’s performance on the given MTL task. Meanwhile, Subspace Network clearly outperforms all three, even DNN equipped with censoring + low-rank, suggesting the advantage of our proposed online one-pass sensing and feed-forward training strategy. The performance advantage of Subspace Network is confirmed across different rank assumptions, and across both synthetic and real data.

---

### Official Review · AnonReviewer3 · 2017-11-28
**This paper introduces a multi-task network architecture within which low-rank parameter spaces were found using matrix factorization. As carefully proved and tested, only one pass of the training data would help recover the parametric subspace, thus network could be easily trained layer-wise and expanded.**

**Rating:** 5
**Confidence:** 4

**Review:**

This paper presents a new multi-task network architecture within which low-rank parameter spaces were found using matrix factorization. As carefully proved and tested, only one pass of the training data would help recover the parametric subspace, thus network could be easily trained layer-wise and expanded.

Some novel contributions:
1. Layer by layer feedforward training process, no back-prop.
2. On-line settings to train parameters ( guaranteed convergence in a single pass of the data)

Weakness :
1. The assumption that a low-rank parameter space exists among tasks rather than original feature spaces is not new and widely used in literature.
2. The proof part(Section 2.2) can be extended with more details in Appendix.
3. In synthetic data experiments (Table1), only small margins could be observed between SN, f-MLP and rf-MLP, and only Layer 1 of SN performs better above all others.
4. Typo: In Table2,3,5, Multi-l_{2,1} (denotes the L2,1 norm) were written wrong.
5. In the synthetic data experiments on comparison with single-task and multi-task models, counter-intuitive results (with larger training data split, ANMSE raises instead of decreases) of multi-task models may need further explanation.
6. Extra models like Deep Networks with/without matrix factorization could be added. ( As proposed model is a deep model, the lack of comparison with deep methods is dubious)
7. In Section 4.2, the real dataset is rather small thus the results on this small dataset were not convincing enough. SN model outperforms the state-of-the-art with only small margin. Extensive experiments could be added.
8. The performance on One-Layer Subspace Network (with only the input features) could be added.

Conclusion:
Though with a quite novel idea on solving multi-task censored regression problem, the experiments conducted on synthetic data and real data are not convincing enough to ensure the contribution of the Subspace Network.

---

> ### Author Response · Authors · 2017-12-26
> **Response for Reviewer 1 (Part 1)**
>
>
> Thanks very much for appreciating the novelty of our work, and for your very insightful and constructive comments. We have addressed all your questions below. In particular, we have significantly enriched our experiments as suggested. All results consistently suggest the performance advantage and robustness of Subspace Network over state-of-art linear /nonlinear methods, as well as DNNs. We have also revised the paper and corrected typos.
>
> Q1: The assumption that a low-rank parameter space exists among tasks rather than original feature spaces is not new.
>
> A1: SN was mainly built on the work series of online subspace sensing, where low-rank assumption was enforced on the input space, e.g., Mardani et al. (2015), Shen et al. (2016). Motivated by the popularity of low-rank parameter space in MTL, we introduced the first-of-its-kind combination of online subspace sensing (mostly focusing on input space), and low-rank parameter assumption for MTL: we believe their marriage to be new.
>
> Q2: The proof part (Section 2.2) can be extended with more details in Appendix.
>
> A2: We’ve extended more detailed proof outlines in Appendix. At some steps of the proofs, we point to the important key result to refer to. Proofs are provided for self-containedness only.
>
> Q3: In synthetic data experiments (Table1), only small margins could be observed between SN, f-MLP and rf-MLP, and only Layer 1 of SN performs better above all others.
>
> A3: We have provided additional experimental results after further tuning all three networks. In addition to subspace difference, we added two new metrics: (1) the maximum mutual coherence of all column pairs from two matrices, as a classical measurement on how correlated the two matrices' column subspaces are; (2) the mean mutual coherence of all column pairs from two matrices. Note that the two mutual coherence-based metrics are more robust since they are immune to linear transformations of subspace coordinates, to which the L2-based subspace difference is not immune to. Results can be found in Table 1, showing the clear advantage of SN over f-MLP and rf-MLP under all three metrics. The performance margins of SN in terms of maximum/mean mutual coherences are remarkably more visible than under L2-based difference.
>
> Q4: In the synthetic data experiments on comparison with single-task and multi-task models, counter-intuitive results (with larger training data split, ANMSE raises instead of decreases) of multi-task models may need further explanation.
>
> A4: Thanks so much for pointing out. We found there were numerical convergence issues with the optimization algorithms for MTL models.  We have fixed the problem and updated the results in Table 3, where the performance turns now intuitive.
>
> We also extended our experiments to compare results between considering target boundedness (Censored) and not (Uncensored) for both single-task and multi-task models. In either scenario, we reported the best performance (LS+L1 for single task linear, and Multi Trace for multi-task in our case) in Table 2.
>
> Q5: Extra models like Deep Networks with/without matrix factorization could be added.
>
> A5: Thanks very much for your suggestion. We have added them as suggested. Please refer to the new paragraph “Benefits of Going deep” in Section 4.1 and “Performance” in Section 4.2 in the revised paper. In sum, we compared with three DNN baselines (naive, with censoring, and with censoring + low-rank) for both synthetic and real data. The comparisons of three baselines indicated that both censored regression and low-rank assumption improved DNN’s performance on the given MTL task. Meanwhile, SN clearly outperformed all three, even DNN equipped with censoring + low-rank, suggesting the advantage of our proposed online one-pass sensing and feed-forward training strategy. The performance advantage of SN over DNNs was confirmed across different rank assumptions, and across both synthetic and real data.

---

> > ### Author Response · Authors · 2017-12-26
> > **Response for Reviewer 1 (Part 2)**
> >
> >
> > Q6. In Section 4.2, the real dataset is rather small thus the results on this small dataset were not convincing enough. Extensive experiments could be added.
> >
> > A6: Thanks for the comments. We have extended our experiments by comparing SN with several baseline DNNs (in previous comments) and results verify that SN outperforms all three variants DNNs.
> >
> > We note that the MTL, like the one proposed in this paper, is typically used for solve learning problems with insufficient training data. Nowadays this is very typical in medical research, and is one motivation for us to design this method. The ADNI data used in our paper is so far the largest cohort collected for Alzheimer’s disease study, despite that it still only has less than 1000 patients available due to the expensive data collection process. The ADNI dataset is widely used for building machine learning models, where researchers proposed many algorithms to tackle challenges arising from the small sample-size:
> >
> > [1] Huang, S., Li, J., Sun, L., Ye, J., Fleisher, A., Wu, T., ... & Alzheimer's Disease NeuroImaging Initiative. (2010). Learning brain connectivity of Alzheimer's disease by sparse inverse covariance estimation. NeuroImage, 50(3), 935-949.
> > [2] Stonnington, C. M., Chu, C., Klöppel, S., Jack, C. R., Ashburner, J., Frackowiak, R. S., & Alzheimer Disease Neuroimaging Initiative. (2010). Predicting clinical scores from magnetic resonance scans in Alzheimer's disease. Neuroimage, 51(4), 1405-1413.
> > [3] Orrù, G., Pettersson-Yeo, W., Marquand, A. F., Sartori, G., & Mechelli, A. (2012). Using support vector machine to identify imaging biomarkers of neurological and psychiatric disease: a critical review. Neuroscience & Biobehavioral Reviews, 36(4), 1140-1152.
> > [4] Zhang, D., Shen, D., & Alzheimer's Disease Neuroimaging Initiative. (2012). Multi-modal MTL for joint prediction of multiple regression and classification variables in Alzheimer's disease. NeuroImage, 59(2), 895-907.
> > [5] Zhou, J., Liu, J., Narayan, V. A., Ye, J., & Alzheimer's Disease Neuroimaging Initiative. (2013). Modeling disease progression via MTL. NeuroImage, 78, 233-248.
> > [6] Liu, M., & Zhang, D. (2016). Pairwise constraint-guided sparse learning for feature selection. IEEE transactions on cybernetics, 46(1), 298-310.
> > [7] Zheng, X., Shi, J., Li, Y., Liu, X., & Zhang, Q. (2016, April). Multi-modality stacked deep polynomial network based feature learning for Alzheimer's disease diagnosis. In Biomedical Imaging (ISBI), 2016 IEEE 13th International Symposium on (pp. 851-854). IEEE.
> >
> > We do plan to collaborate with clinical partners to collect larger neurodegenerative datasets and apply the proposed method.
> >
> > Q7. Add 1-Layer SN results
> >
> > A7: The results of 1-layer SN have been added: see Table 2 and Table 5.

---

### Author Response · Authors · 2017-12-26
**Overall Response**

We would like to thank all reviewers for all the valuable and constructive comments. By following the suggestions, we have significantly extended our experiments, revised our paper and addressed questions from all the reviewers. Below we summarize the improvements in this revision:

Major Improvements:

1. For synthetic experiments (Table 1), we provide additional metrics to evaluate the recovery achieved by different methods. Results show that SN outperforms all methods in various metrics, with comparable margins.

2. We largely extend experiments and compare SN with three DNN baselines (naive, with censoring, and with censoring + low-rank) for both synthetic and real data. Results (Table 2 and 5) show SN outperforms all baselines.

3. We verify the benefit of considering target boundedness in all sets of methods considered: (1) Single-task and multi-task shallow methods; (2) Deep methods. Results (Table 2 and 5) show performance improvement when considering the boundedness of targets.

4.  We’ve extended more detailed proof outlines for convergence properties for both asymptotic and non-asymptotic cases in Appendix.

Minor Improvements:

1. We solve the counter-intuitive experimental observation by fixing numerical issues in multi-task algorithms and update results in Table 2 and 5.

2. We remeasure the computation speed more accurately and update results in Table 6.

3. For all the confusions we made previously, we have made it more clear in the revised version.

---

### Author Response · Authors · 2018-01-04
**Warm Reminder**

We wish all the reviewers happy new year and we are looking forward to addressing any new comments to our responses posted previously. If you have any comments, feel free to let us know. Thank you very much.

---

### Decision · Program_Chairs · 2018-01-29
**ICLR 2018 Conference Acceptance Decision**

**Decision:**

Reject

**Comment:**

Authors present a method for modeling neurodegenerative diseases using a multitask learning framework that considers "censored regression" problems (to model where the outputs have discrete values and ranges). Given the pros/cons, the committee feels this paper is not ready for acceptance in its current state.


Pro:
- This approach to modeling discrete regression problems is interesting and may hold potential, but the evaluation is not in a state where strong meaningful conclusions can be made.

Con:
- Reviewers raise multiple concerns regarding evaluation and comparison standards for tasks. While authors have added some model comparisons in response, in other areas comparisons don't appear complete. For example, when using MRI data, networks compared all use features derived from images, rather than systems that may learn from images themselves. Authors claim dataset is too small to learn directly from pixels in this data (in comments), but transfer learning and data augmentation have been successfully applied to learn from datasets of this size. In addition, new multitask techniques in the imaging domain have also been presented that dynamically learn the network structure, rather than relying on a hand-crafted neural network design. How this approach would compare is not addressed.